# Analysis of the Characteristics and Causes of Night Tourism Accidents in China Based on SNA and QAP Methods

**DOI:** 10.3390/ijerph20032584

**Published:** 2023-01-31

**Authors:** Rui Huang, Chaowu Xie, Feifei Lai, Xiang Li, Gaoyang Wu, Ian Phau

**Affiliations:** 1College of Tourism, Huaqiao University, Quanzhou 362021, China; 2Tourism and Aviation Service College, Guizhou Minzu University, Guiyang 550025, China; 3School of Management and Marketing, Curtin University, Perth, WA 6845, Australia

**Keywords:** night tourism, safety accidents, accident characteristics, risk inducing factors, social network analysis, quadratic assignment procedure

## Abstract

The key purpose of this paper is to address an inherent gap in the literature on safety issues in the development of night tourism. This research takes a novel methodological approach, by using 8787 cases of tourism safety accidents in typical night tourism cities in China, and applying social network analysis (SNA) and quadratic assignment procedure (QAP) regression analysis to explore the multidimensional structural characteristics and risk-causing factors of night tourism accidents. Key findings include: (1) Amidst the complexity and diversity of the night tourism safety accidents in cities, disastrous accidents, public health accidents, natural disasters, and social security accidents are the main types of night tourism safety accidents. (2) Night tourism safety accidents have strong aggregation in specific time periods and spatial regions. There are differences in the timepoint and duration of each accident type, showing different distribution characteristics in different cities and locations. (3) Distribution of accident types in night tourism products shows obvious core-edge structure characteristics. (4) The degree of co-occurrence of four risk-inducing factors, i.e., personnel, facilities, environment, and management, has high explanatory power at the accident correlation level in the co-occurrence network of night tourism safety accidents in cities, and the influence effects of risk factors are heterogeneous at different timepoints. Our results provide some valuable implications for optimizing night tourism safety governance in cities.

## 1. Introduction

The extant literature has shown that night tourism is a growing global phenomenon and is an important financial and cultural contributor to the night economy [1]. It is widely known to have a better leisure atmosphere than daytime tourism experience activities [2]. Specifically, in China, the central government is urging major cities to accelerate the establishment of a “dual circulation” development pattern aimed at prioritizing domestic consumption while remaining open to international investment and trade [3]. Night tourism is regarded as a “booster” for expanding domestic demand, promoting consumption, and creating employment, which are of great significance for deepening supply-side structural reform, cultivating new driving forces of urban development, and promoting high-quality development of national economy [1,4,5].

Night tourism has recently had a breakthrough in reviving the domestic consumer market and boosting the tourism industry economy. Local governments have introduced relevant policies and measures, such as supporting night markets, extending the operating hours of retailers, and investing in landscape lighting to stimulate night consumption [6]. 

However, there are still some issues in the early development of night tourism ranging from inadequate use of lighting, congestion of space and traffic, poor service quality, and product homogeneity, which are causing a slightly negative impact on the tourist experience [7,8]. More importantly, the sustainable development of night tourism in Chinese cities is plagued by aggravating safety issues. According to the travel agency liability insurance data from the Ministry of Culture and Tourism of China, 28,800 night tourism safety accidents occurred in tourism cities across the country from 2010 to 2019, and the number of tourist casualties reached 41,200 [9]. Safety after dark is a primary concern of tourists and may drive risk-averse tourists away as tourism accidents have a profound effect on the image of a city’s tourism industry, which impacts the choice of tourist destinations by potential tourists [10,11,12]. 

Current research on night tourism has mainly focused on landscape construction [13], atmosphere creation [1], product experience [4], tourists’ perceived value [12], as well as travel motivation and satisfaction [14]. Notwithstanding the critical role that safety plays in night tourism, safety issues during nighttime travel have received scant attention. The unique time period of night tourism raises a few issues; the rapid decline of tourists’ alertness and physical function, coupled with the complexity of urban environmental risks and inadequate response of urban safety management, have led to more frequent tourism safety accidents in night tourism than in daytime tourism [15]. In addition, the development of night tourism challenges the social order and cultural atmosphere of a city, risking community organizations, collective lifestyles, and individual citizen behavior [13]. Undoubtedly, safety accidents are more rampant in cities where night tourism already exists. Researchers have used qualitative analysis approaches to evaluate individual issues such as sexual crime, street violence, drug abuse, alcoholism, and robbery [5,16,17,18]. To the best of our knowledge, few studies have systematically discussed safety issues of night tourism across a range of issues based on large-scale tourist accident data. Further, there is a lack of systematic quantitative empirical analysis on the structural characteristics and causes of night tourism safety accidents from an overarching perspective. 

To fill this gap, this research takes a novel methodological approach, by using 8787 cases of tourism safety accidents in typical night tourism cities in China, and applying social network analysis (SNA) and quadratic assignment procedure (QAP) regression analysis to explore the multidimensional structural characteristics and risk-causing factors of night tourism accidents (Figure 1). This research makes the following contributions. Firstly, we establish the relationship network among related elements of accidents, such as the type, time of occurrence, site location, and geographical region, which reveals the common characteristics of night tourism accidents in different cities. Secondly, based on the 4M theory framework, we precisely and comprehensively identify 16 subrisk factors and 4 major risk factors that affect night tourism accidents. We also examine the effects of personnel, facilities, environment, and management on tourism safety accidents at different time periods with the help of QAP analysis and SNA. The key purpose of this paper is to provide a theoretical reference for enhancing night tourism safety governance in cities. The remainder of this paper proceeds as follows: In the second section, we review the relevant literature; in the third section, we describe the data processing and method application; in the fourth section, we present the analytical process and discussion; in the fifth section, we present conclusions and implications.

## 2. Relevant Literature

### 2.1. Night Tourism Defined

Night tourism is defined as any type of tourism activity that occurs between 6 pm and 6 am, and that is an extension and expansion of regular tourism activities from the daytime [4,12,19]. 

The development of night tourism enables tourists to fully understand unique tourist destinations and to experience the real atmosphere that cannot be felt during daytime. Relevant research has explored night tourism from the aspects of perceived value [12], tourists’ motivation and satisfaction [14], tourists’ willingness to pay and revisit intention [1,4], and destination attachment [12]. Night tourists can visit night scenic spots [15], experience gastronomic night market food [20], watch night performances [21], experience night tourism culture [4], and enjoy rich tourism experiences [4,22]. 

Relevant studies on tourist destinations have found that night tourism is one of the most important means to enhance the economic vitality of cities and to promote the sustainable development of urban tourism [4]. The former, i.e., economic vitality, informs the market consumption demand of tourists, increases potential business opportunities and urban employment opportunities, and drives 24-h economic growth of cities [23,24]. The latter, i.e., sustainable development, promotes the development and protection of urban night scenic spots [13], expands the construction of urban leisure and cultural places, enriches local culture, coordinates the host–guest relationship, and further promotes the dissemination of urban culture [25,26]. Other studies have also focused on the development of urban cultural products [4,21,27], urban landscape construction of night tourism [13], destination atmosphere authenticity [20], destination image management [28], and the development of countermeasures of night tourism [29]. 

### 2.2. Night Tourism Safety Risks

Night tourism safety, which is a basic condition for successful development of night tourism in cities, has its challenges, in particular, its unique risks [13]. First and foremost, maintaining public order and safety control of social activities is hampered by poor visibility, especially when tourists are less vigilant due to the physical and mental fatigue of the day, and therefore, more vulnerable to hidden risks [13,19,24]. The large-scale expansion of night tourism activities and the leisure and entertainment industry have led to urban violence and chaos, including conflict incidents between residents and tourists and between hosts and guests [25,30,31,32]. As compared with commuters who regularly and consistently travel between cities and places of residence, tourists are often in completely unfamiliar environments [33,34], which means that the behavioral pattern of night tourists are often met with higher risks [35,36,37]. 

Second, high-risk behaviors of young tourists in bars, nightclubs, and other night spots including excessive drinking, recreational drug abuse, and casual sex [38,39], have increased the possibility of illness, injury, and sudden/unnatural death. Related studies have also found that this has led to a sharp rise in crime rates, street violence, sexual crime, theft, and robberies [5,16,17,18,40]

In addition, there have been calls for many other related issues to be resolved including risky parking issues and traffic safety [17]; food safety issues, in particular, in night markets [14]; the risk of sexual harassment of the young women travelers during the night [19]; and risks of overtourism in the urban night economy [8]. 

### 2.3. Destination Tourism Safety Accidents

A destination tourism safety accident refers to an unexpected event that violates a tourist’s will, forces the suspension of travel, and causes serious personal injury and property loss to the tourist [41]. Tourism safety incidents reflect the risk situations of tourist destinations, and it is important to explore the pattern of tourism safety accidents in the destination, for better risk prevention and control. Existing studies have analyzed the destination tourism safety accidents from three aspects namely: 

(a) Diversity and complexity in the types of accidents including food poisoning [8], accommodation theft [42], traffic accidents [43], entertainment and amusement ride accidents [44], and customer fraud [45]. 

(b) Temporal and spatial distribution of accidents including monthly and regional distribution of overseas visitor injuries [10], record of seasons and cities of brain injury accidents among Austrian tourists [46], and spatial distribution pattern of tourist robberies in American urban tourist attractions. Interestingly, many of these have occurred in scenic spots with very little accessibility of police support [47]. 

(c) Factors causing the accidents in accordance to the “4M” accident-causing theory [10,48,49] which are (i) dangerous behavior of humans, such as unsafe and reckless driving behaviors causing traffic accidents [43]; (ii) unsafe condition of facilities, such as poor destination infrastructure and tourism reception facilities and equipment [15,50]; (iii) risks in the environment, such as the impact of winter weather conditions on Finnish tourist accidents [51]; (iv) insufficient management measures, such as poor and slow emergency response in highly aggregated tourist crowds [52]. 

### 2.4. Gaps in the Literature

Night tourism accidents are not rare in China’s urban tourism industry and have resulted in a large number of fatalities and substantial property damage. However, to date, far too little attention has been paid to night tourism accidents. First, most existing studies have been qualitative and descriptive in nature and have only focused on single risk types of night tourism. What is inherently deficient is an overarching approach that summarizes the structural characteristics of night tourism safety accidents and compares different cities. The SNA method is based on the co-occurrence relationship of nodes as the examination element, and it is suitable for revealing the overall characteristics of night tourism accidents via relational data. This paper serves to close this gap by applying a social network analysis approach with a unique classification across typical night tourism cities in China as case sites to provide a theoretical reference for optimizing night tourism safety governance in cities. Second, there are few studies on the influencing factors of night tourism accidents based on large-scale data, and there is a lack of relevant theoretical support. There is a need to propose a theoretical framework to establish the correlation between risk factors and night tourism accidents, and to compare the impacts of various risk factors on the accidents with the help of the QAP in SNA. Once we have a better understanding of the causes of night tourism accidents, corresponding measures can be adopted to improve the safety of tourists.

## 3. Research Design

### 3.1. Data Source and Processing

The data source for analysis in this paper is derived from the Chinese nation-wide platform of Unified insurance demonstration project of travel agency liability insurance. This travel agency liability insurance is a type of insurance coverage that the government mandates for travel agencies to insure for tourists. According to the statistics of the Ministry of Culture and Tourism, more than 20,000 travel agencies have participated in the unified insurance demonstration project by 2020 with a coverage of more than 80%. This data source provides the advantage of a large sample size and information records of high authenticity and reliability. The unit of analysis for this study is the cities with well-developed night tourism destinations as these cities have high occurrences of night tourism safety accidents. The sample of 40 cities selected for this study was comprised of (a) the top 20 typical nighttime economic cities in China selected from the “China Night-time Economy Development Report” (2020) [6] issued by the China Tourism Academy, and (b) 20 typical leisure tourism cities selected from the “China Urban Leisure and Tourism Competitiveness Report” (2020) [53]. In total, 8787 tourism safety accident cases extracted from the platform were from 2010 to 2019 and occurred between the time frame from 18.00 to 06.00 h on the following day. The samples were deconstructed and numerically coded with basic variables such as the specific timepoint of occurrence, accident type and cause, and type of tourism product. Finally, the accident analysis data were converted into a collinear matrix, and then imported into the UCINET and ArcGIS10.3 software for analysis.

### 3.2. Method

In recent years, social network analysis (SNA) has been increasingly used in accident case analysis of safety science [54,55,56]. By establishing a connection matrix, the relationship between system elements could be explored. With the help of network visual analysis technology, the complex and diverse elements in accident cases could be clearly presented. The network diagram of safety accident types and degree of co-occurrence of various factors in night tourism of different cities was constructed, and the characteristics of correlation structure between different accident factors and accident types were explored by using the UCINET software. 

In the network structure diagram, each accident element node represents a new research object, while the connecting line represents the relationship between elements. The thickness of the line is directly proportional to the degree of element association, and the node degree centrality is used to measure the feature of element relationship. Degree centrality is a reflection of the importance of the node location in the whole network [57]. The higher its value, the more relationships a node has in the whole network. The core-periphery structure is a network structure in which the nodes in the middle core area are closely connected and the nodes in the outer edge area are loosely connected [58]. The core-periphery analysis could help to quantify the degree of connection between nodes, and effectively distinguish the types of core accidents and marginal accidents in night tourism; QAP is a method to compare the similarity of elements in the square matrix [59] and a QAP regression analysis is often used to investigate the relationship between multiple matrices and single matrix [60]. 

## 4. Analysis and Discussion 

### 4.1. Types of Night Tourism Accidents

According to the classification criteria of safety accidents determined in the Law of the People’s Republic of China on Emergency Response, night tourism safety accidents in the cities are divided into four main types, i.e., disastrous accidents, natural disasters, public health accidents, and social security accidents; these main types are subdivided into 14 accident subtypes, and subsequently, 40 accident basic types. The types of night tourism safety accidents in cities are presented in Table 1. The incidence rate of disastrous accidents is the highest (53.19%) and the majority of this group includes fall and slip accidents, road traffic accidents, and infrastructure accidents. Among public health accidents (28.81%), food poisoning accounts for the highest proportion. Among the natural disaster accidents (8.95%), the majority of cases are meteorological disasters such as typhoons, rainstorms, and lightning strikes. Among social security accidents (9.05%), the main basic types are theft, robbery, and fraud. 

### 4.2. Temporal Distribution of Night Tourism Accidents

#### 4.2.1. Time Distribution of Night Tourism Accidents

As presented in Figure 2, there is a pattern of a decreasing number of safety accidents from 18:00 until 06:00 the next morning. In particular, the peaks are at 18:00 and 20:00, probably because most of night tourism activities are concentrated during this time period. There is a dip from 21:00 to 00:00, probably because some night tourism activities in cities have gradually ended. Night tourism accidents start to rise again from 00:00 to 02:00, where the most tourist sudden illness accidents occur in this period. The lowest count in the number of accidents is from 02:00 to 04:00, when most of the night tourism activities in cities have ended. 

#### 4.2.2. Time Distribution Considering Different Accident Types

Figure 3 presents the radar structure diagrams which identify the temporal distributions of different accident types. Among disastrous accidents, the accident subtypes mainly occurs between 18:00 and 23:00. Traffic accidents and facilities and equipment accidents reach a peak at 18:00, while crowd gathering accidents reach a peak at 19:00. Accidental injuries and fire safety accidents reach a peak at 20:00 and 22:00, respectively. Among the public health accidents, night food poisoning accidents are mainly concentrated in the period from 18:00 to 20:00. The cases of sudden disease mainly occur from 18:00 to 20:00, the cases of cold and fever of tourists mainly occur from 22:00 to 23:00 when the temperature drops, and cases of sudden death while sleeping mainly occur from 01:00 to 03:00 in the morning. Among natural disaster accidents, meteorological disasters that occur at 18:00 are obviously higher than those at other timepoints. In fact, these accidents such as typhoon disasters, heavy rain, and lightning could affect the night activities planned during the night or directly lead to cancellations or delays. Among social security accidents, thefts, fraud, and public security crimes mainly occur from 18:00 to 21:00. It is worth noting that the number of public security crimes at 02:00 is also high, mainly from robbery and sexual assault.

### 4.3. Spatial Distribution Characteristics of Night Tourism Safety Accidents

#### 4.3.1. Regional Distribution Considering Different Accident Types

Figure 4 shows a regional distribution map that reveals the spatial distribution of different accident types. Overall, night tourism safety accidents show heterogeneous distributions in various cities in China. All types of night tourism safety accidents occur more predominantly in Lijiang, Kunming, and Sanya. Public health accidents are more common in coastal cities such as Dalian, Qingdao, and Xiamen. Disastrous accidents occur more frequently in Guilin, Huangshan, Hangzhou, and Zhangjiajie. Social security accidents are more common in Changsha, Guilin, Zhangjiajie, Xiamen, and Xi’an. Natural disaster accidents are more frequent in Shenzhen, Guangzhou, Xiamen, Hangzhou, and Shanghai. 

In order to further explore the relationships between accident subtypes and cities, a two-mode structure network between safety accident subtypes and cities is established based on the UCINET software. The network centrality of each city node is calculated as reflected in Table 2. The network structure diagram of cities and accident subtype nodes was drawn with Netdraw plug-in, and is presented in Figure 4. 

Some interesting observations emerged from this analysis. First, the structure network between disastrous accidents and cities (Figure 5a) shows that Sanya, Kunming, Lijiang, and Beijing have a large degree of centrality, which is closely related to each accident subtype. In terms of node connection intensity, shipping accidents are highly correlated with coastal tourist cities such as Sanya, Xiamen, and Qingdao. Infrastructure accidents and entertainment facilities accidents are highly correlated with cities such as Kunming, Lijiang, and Sanya. Fall and slip accidents are highly correlated with scenic tourist cities such as Huangshan, Guilin, and Zhangjiajie. Stampede accidents and missing tourists are highly correlated with mega tourist cities such as Shanghai, Beijing, and Shenzhen. Animal attacks are highly correlated with Sanya, Huangshan, and Guilin. The correlations between explosion and electric shock accidents and cities are low and random.

Second, the structure network between public health accidents and cities (Figure 5b) shows that Kunming, Lijiang, Guangzhou, Guiyang, and Lhasa have a large degree of point centrality, which presents complex correlation characteristics with various accident subtypes. In terms of node connection strength, food poisoning has a high correlation with the coastal tourist cities of Kunming, Lijiang, Dalian, Sanya, and Xiamen. Cases of cold, fever, sudden death, and sudden disease have high correlations with cities in northern China such as Dalian and Harbin. Altitude sickness has a high correlation with Lhasa, Lijiang, and Kunming. Heat stroke has a high correlation with Chongqing, Changsha, and Wuhan.

Third, the structure network between natural disaster accidents and cities (Figure 5c) shows that Hangzhou, Wuhan, Chengdu, Dalian, and Kunming have a large degree of point centrality, but the types of natural disaster accidents associated with such cities are complex. In terms of node connection strength, rainstorm, lightning, and typhoon disasters are highly correlated with coastal tourist cities such as Sanya, Xiamen, and Guangzhou. Urban waterlogging is highly correlated with mega cities such as Beijing and Shanghai, as well as historical and cultural tourist cities such as Wuhan, Changsha, and Hangzhou. Ice and snow disasters are highly correlated with Harbin, Shenyang, and Changchun, which are northern cities in China dominated by ice and snow tourism. Extreme high temperatures are highly correlated with cities such as Sanya, Xiamen, Chongqing, and Changsha. Haze and dust are strongly correlated with Beijing, Zhengzhou, and Lanzhou.

Finally, the structure network between social security accidents and cities (Figure 5d) shows that Kunming, Lijiang, and Dalian have a large degree of point centrality; similarly, the structure of social security accidents in these cities is relatively complex. In terms of node connection strength, public security crimes such as theft, robbery, and fraud are highly correlated with cities such as Lijiang, Kunming, and Sanya. Violent conflicts such as fighting and drinking are highly correlated with northern cities such as Shenyang, Changchun, Harbin, and Dalian.

#### 4.3.2. Site Distribution Considering Different Accident Types

In this study, we analyzed the distribution of night tourism safety accidents in the specific categories and sites of the cities as presented in Table 3. The highest category for accidents is at entertainment places which accounts for 30.8%, and theme park sites have the highest frequency at 9.35%. This is followed by the category of places for sightseeing, which accounts for 24.34% and ancient streets and towns sites have the highest frequency of accidents of 6.67%. The category of catering places accounts for 15.90% of accidents, and food stalls sites have the highest frequency of 6.07%. The categories of accommodation, shopping, and traffic places account for 11.75%, 9.28%, and 7.90% of accidents, respectively. In order to explore the site distribution characteristics of different accident types, a two-mode network between sites and accident types is established. The network centrality and network structure are presented in Table 4 and Figure 6.

Some interesting observations emerged. First, disastrous accidents, scenic attractions, ancient streets and towns, gourmet plazas, and commercial pedestrian streets have high point centrality. In terms of node connection intensity, road traffic accidents and shipping accidents are strongly related to sidewalks, traffic-ways, and seashores, respectively. Infrastructure accidents are related to catering and accommodation sites such as hotels, B&Bs, and food stalls. Entertainment facility accidents are closely related to cultural and creative blocks, theme parks, entertainment plazas, and public gardens. Crowd gathering accidents and lost tourist accidents are related to ancient streets and towns, and scenic attractions. Fire, explosion, and electric shocks are strongly related to theatre/show sites, KTV/dance hall. sites, and food stalls. Fall and slip accidents are related to places for sightseeing and entertainment, such as theme parks, scenic attractions, scenic belts, ancient streets and towns, and theater and show sites. 

Second, in public health accidents, catering and accommodation places such as hotels, B&Bs, food stalls, and bars have high point centrality, and there are many types of public health accidents associated with such places. In terms of node connection strength, food poisoning is strongly related to dining rooms, gourmet plazas, and food stalls. The viral epidemic is closely related to commercial pedestrian streets and shopping malls. Sudden disease and sudden death are closely related to hotels, B&Bs, theme parks, health clubs, and KTV/dance halls. The cases of cold and fever, and heatstroke are closely related to seashores, scenic belts, and theme parks.

Third, in natural disaster accidents, scenic attractions have high centrality, followed by ancient streets and towns. In terms of node connection strength, typhoon disaster is strongly related to seashores, ancient streets and towns, and scenic belts. Thunderstorm disasters are strongly related to band shells and commercial pedestrian streets. Ice and snow disasters are strongly related to theme parks, traffic-ways, and stations. 

Finally, in social security accidents, crowded places, such as bars, food stalls, gourmet plazas, and commercial pedestrian streets have high centrality, and the distribution of social security accidents in such places is complex. In terms of node connection strength, theft is strongly related to hotels, B&Bs, and commercial pedestrian streets. Fraud is strongly related to shopping malls, handicraft shops, game rooms, and catering places. Robbery is strongly related to ancient streets and towns, sidewalks, and commercial pedestrian streets. Drunk/aggression and fights are strongly related to bars and food stalls.

### 4.4. Classification of Accident Types in Different Product Projects

According to the existing literature [22,27], this study divides night tourism into seven categories namely commercial street strolling (12.34%), performing arts (7.19%), folk festivals (12.48%), leisure and wellness (8.71%), city sightseeing (21.09%), authentic food experiences (22.85%), and entertainment experiences (15.34%). 

In order to identify the classification of accident types in different product projects, a core-edge analysis of SNA method was conducted. As presented in Figure 7, the fitting coefficient shows that the core-edge structure of the performing arts enjoying is the most obvious. Lost tourist accidents, stampede accidents, fire accidents, infrastructure accidents, and fall and slip accidents are the core accident types, which mainly occur in theater viewing, lighting facilities, fireworks performances, live performances, and other project activities

The fitting coefficient and core density of folk festival experience are 0.672 and 0.648 respectively. The types of accidents in the core area are obvious, including stampede accident, fall and slip accident, fire and explosion, which mainly happened when tourists participate in festivals such as folk festival experiences, music festivals, art festivals and ice and snow festivals. The fitting coefficient and core density of leisure and wellness are also relatively high. The types of accidents in the core area are sudden death, sudden disease, skin sensitivity and accidental burns, which mainly happened when tourists take part in urban recreation activities such as bathing sauna and SPA massage. The fitting coefficient and core density of authentic food experiences are 0.351 and 0.436 respectively. The types of accidents in the core area are food poisoning, scald, drunk and aggression accidents, fights, and infrastructure accidents. The fitting coefficient and core density of the sightseeing are 0.115 and 0.339, respectively, and the number of core accident nodes in the night tourism product projects is complex, which mostly happened in the process of tourists’ night bund tours, water scenery belts, ancient streets and towns, and scenic attractions. The fitting coefficient and core density of commercial street strolling is low. As the activities of commercial street strolling involve diversified tourism elements such as leisure, entertainment, catering, and shopping, the distribution of tourism safety accidents is relatively homogeneous, and the core-edge structure is not obvious. The fitting coefficient and core density of entertainment experience is the lowest. Due to the small difference in the degree of co-occurrence of each accident type in entertainment experience projects, it is not easy to form an obvious stratification, therefore, it does not show a core-edge structure.

### 4.5. Analysis on the Causes of Night Tourism Safety Accidents

#### 4.5.1. Risk Factors of Nighttime Tourism Accidents

The 4M theory (man-machine-media-management) is widely used as a typical theoretical model in the field of safety science research, such as engineering management [61], coal mining [62], marine transportation [63], and construction projects [64]. In addition, there are also studies that have applied it to urban public safety risk analysis [65]. In tourism, Bentley et al. (2001) [10] used the 4M theory to investigate the risk factors that led to injury for adventure tourists in New Zealand [10]. Xie et al. (2021) [49] identified the tourist safety perception structure at destinations based on this theory. Thus, it is appropriate to introduce teh 4M theory to explain the causes of safety accidents in night tourism. The 4M theory indicates that the influencing factors of accidents involve four elements, including dangerous behavior of humans, unsafe condition of facilities, risks in the environment, and insufficient management measures [10,49]. Based on this theoretical framework, the 8787 insurance accident case causes of night tourism accidents are divided into four structural risk factors containing personnel risk, facility and equipment risk, environmental risk, and management risk (Figure 8).

In this study, we decomposed and extracted the cause details of each accident. Then, four risk structure factors were classified and counted in this research (see Table 5). The results show that the total proportion of personnel risk factors is 33.97%. The individual factors of tourists are the main reasons, with a cumulative proportion of 27.08%, including tourists’ poor physical quality (16.25%) and weak safety awareness (10.83%). The management risk factors account for 24.29%, of which the public health management risk is the highest, accounting for 9.40%, and the risk factors of market supervision and management and social security management are 6.89% and 6.07%, respectively. The proportion of environmental risk factors reach 24.16%, among which atmospheric environmental risk accounts for the highest proportion, followed by road environmental risk and tourism environmental risk, and geological environmental risk accounts for the lowest proportion. The proportion of facility and equipment risk factors reach 17.58%, among which the risk proportion of engineering facilities is the highest. Entertainment facility and equipment risk account for 5.95%, while lighting risk and extinguishing/protection risk account for 2.79% and 1.38%, respectively.

#### 4.5.2. QAP Regression Analysis

In order to explore the influence of urban risk-causing factors of night tourism safety on the distribution of urban accidents, the co-occurrence network of risk-causing factors and the co-occurrence network of night tourism safety accidents in cities were established, respectively. In this study, a QAP regression analysis was used to analyze the influence of the co-occurrence of urban personnel risk, facility and equipment risk, environmental risk, and management risk on the network matrix of night tourism safety accidents in the cities, and the observation time periods were further subdivided into 18:00–20:00, 21:00–23:00, 00:00–02:00, and 03:00–06:00 to investigate the differences of various risk factors in different time periods. The QAP regression results are shown in Table 6.

The results show that the adjusted R^2^ is between 0.768 and 0.977, indicating that the co-occurrence of risk factors has a high explanatory power on the correlation level of night tourism safety accident network. Overall, from 18:00 to 06:00, the co-occurrence of risk factors has a significant positive impact on the network matrix of night tourism safety accidents. The standardized regression coefficients of the co-occurrence of facility and management risk factors are larger, which are 0.344 and 0.315, respectively (*p* < 0.001).

From 18:00 to 20:00, the standardized regression coefficients of the degree of co-occurrence of personnel risk, facility and equipment risk, environmental risk, and management risk factors have little difference. The reason might be that most of tourists travel during this time period, and multiple potential risks erupt at the same time.

From 21:00 to 23:00, the co-occurrence degree of facility and equipment risk factors has a great impact on the network matrix of night tourism accidents. The reason might be that various functional facilities in the city have been gradually shut down during this period, while some tourists’ tourism activities have not ended, and the weakening of the security level of various infrastructures such as lighting has led to the occurrence of night tourism accidents.

From 00:00 to 00:20, the standardized estimation coefficient of the impact of the co-occurrence degree of personnel risk factors on the network matrix of night tourism accidents is large, followed by the co-occurrence degree of management risk factors, while the estimation coefficient of co-occurrence degree of facility risk factors is not significant. It can be seen that the night tourism accident network is affected by the co-occurrence of personnel and management risk factors during this period. This result can be explained by the fact that when the urban temperature drops to a low point in the early morning, the risk of tourists’ physical conditions is high, especially after a long night tour, it is easy to cause cardiovascular issues and sudden death. Further, the density of urban police patrol and the intensity of public safety supervision have also decreased during this period, while the risk factors of social security management and public health management in various cities have increased, which have a greater impact on night tourism safety of the cities.

From 03:00 to 06:00, the standardized estimation coefficient of the impact of the co-occurrence degree of personnel risk factors on the night tourism safety accident matrix reaches 0.453, indicating that the co-occurrence degree of urban nighttime personnel risk factors is an important reason affecting the urban accident network matrix during the period. The estimation coefficient of the co-occurrence degree of management risk factors is the lowest. The reason might be that most tourists have returned to the hotel for rest during this period, and their management risk factors are low.

## 5. Conclusions and Implications

### 5.1. Conclusions

A general description of the characteristics of night accidents based on statistical data was analyzed, and then the impact of risk factors was examined. Some statistical conclusions that are useful for preventing night tourism accidents are summarized as follows:

The types of night tourism accidents in the cities show complex and diversified classification characteristics, including a three-level classification structure. Disastrous accidents, public health accidents, natural disasters, and social security accidents are the four main types. The 14 accident subtypes include traffic accidents, food safety accidents, meteorological disaster accidents, and public security crime accidents. In addition, 40 basic types include stampede accidents, sudden diseases, urban waterlogging, and fraud. Among them, fall and slip accidents, food poisoning, typhoon disasters, and theft accidents account for a relatively high proportion.

Aggregation and heterogeneity exist in the temporal and spatial distribution of night tourism safety accidents in the cities. In terms of temporal distribution, night tourism safety accidents mainly occur from 18:00 to 22:00, and the types of night tourism accidents in these typical cities are different in temporal distribution and duration. In terms of regional distribution, night tourism accidents are mainly concentrated in Kunming, Lijiang, and Sanya. The different types of night tourism safety accidents in different regions have characteristics with clustering and locality. Moreover, the accidents are concentrated in the sites for entertainment or sightseeing, and there are significant differences in the correlation strength between different accident types and urban tourism sites.

Structural differences exist in accident types in night tourism product projects. Among the seven categories of night tourism products, the accidents in four categories of products have an obvious core-edge structure. The core accidents of performing arts enjoying products are lost tourist accidents, stampede accidents, and fire accidents. The core accidents of folk festival products are fall and slip accidents, explosion accidents, and stampede accidents. The core accidents of leisure and wellness products are sudden death, sudden disease, and skin sensitivity. The core accidents of authentic food experience products are food poisoning, and scald, drunk, and aggression accidents.

The risk-inducing factors of night tourism accidents are divided into four structural aspects: personnel risk, facilities and equipment risk, environmental risk, and management risk. The QAP regression analysis shows that the co-occurrence of the four risk-causing factors has a high explanatory power to the accident correlation level in the network of the night tourism accidents. Among them, the co-occurrence of facility and equipment risk and management risk are the main causes that affect the co-occurrence network of urban tourism safety accidents. In addition, the explanatory power of risk factors on night tourism accidents are different at different time periods.

### 5.2. Theoretical Implications

The theoretical implications of our study are mainly reflected in two aspects. First, there is no doubt that safety plays a critical role in the development of night tourism, but the safety issues of night tourism have received little attention. The limited studies on night tourism safety accidents have been mostly restricted to a single city or a single night accident type such as crime accidents, food poisoning accidents, and traffic accidents [14,16,17,39,66]. However, few studies have been able to systematically extract and summarize the common characteristics of tourism safety accidents in different cities. To the best of our knowledge, this is the first study based on large-scale insurance data to comprehensively reveal the multidimensional structural characteristics of night tourism accidents, including spatial-temporal distribution characteristics, type characteristics, and product classification characteristics, which provide the basis for understanding the common sense of night tourism accidents. Second, the occurrence of night tourism safety accidents is influenced by complex and dynamic risk factors. Some studies have explored single influencing factors of night tourism accidents, such as abusive consumption of alcohol, overcrowding, and street lighting conditions [19,39,40]. However, there are gaps that remain in the systematic analysis of which factors affect night tourism safety accidents based on a specific theoretical framework. As a key contribution to the advancement of theory, this study identified the diverse risk factors leading to night tourism accidents rooted in the 4M theory. Although the 4M theory has been widely recognized and applied in the fields of safety science research [61,62,64], few studies have used it to explain the phenomenon of night tourism safety accidents. Our research further expands the theoretical application scenario of the 4M theory.

### 5.3. Managerial Implications

There is a need to strengthen the cooperation between government departments and to enhance the ability of collaborative governance, which involves joint action of local meteorological departments, transportation departments, public security departments, and fire control departments. Local governments need to build a comprehensive emergency response platform to control night tourism safety accidents in cities, to strengthen inter-departmental cooperation and information sharing, and to form a governance pattern of night tourism safety with unified command, coordinated operation, and smooth communication.

There should be a focus on offering a differentiated management and control countermeasure, based on the unbalanced spatial-temporal distribution of accidents. There is a need to strengthen and providing safety assurance resources in accident prone periods and sites, and to strengthen the prevention and control of high-risk spatial-temporal environmental risks of cities. Corresponding security management mechanisms and emergency response plans should be matched according to the occurrence characteristics of the accident types in different spatial-temporal occurrence situations of accidents.

Plans must be implemented to strictly supervise the market of product projects and to strengthen the safety control of high-incidence accident types. Market supervision departments should increase the safety inspection and risk assessment of night tourism product projects in cities, effectively identify the potential risk areas for each product project, and establish an archive information database of night tourism product project accidents.

More importantly, there is a need to establish a systematic and normalized safety governance system based on multiple risk factors. First, the risk of facilities and management should be the focus of risk prevention and control of night tourism safety, the government’s nighttime public safety control ability should be strengthened, and the level of urban infrastructure guarantee should be improved. Second, the publicity and education of tourists’ night travel safety should be strengthened, employees’ safety service skills should be properly trained, and conflict incidents between tourists and urban residents should be effectively managed. Third, the monitoring, early warning, and risk warning of various environmental risk factors should be strengthened. In addition, there is a need to focus on intervention and guidance for the leading risk factors of night tourism at different time periods.

## Figures and Tables

**Figure 1 ijerph-20-02584-f001:**
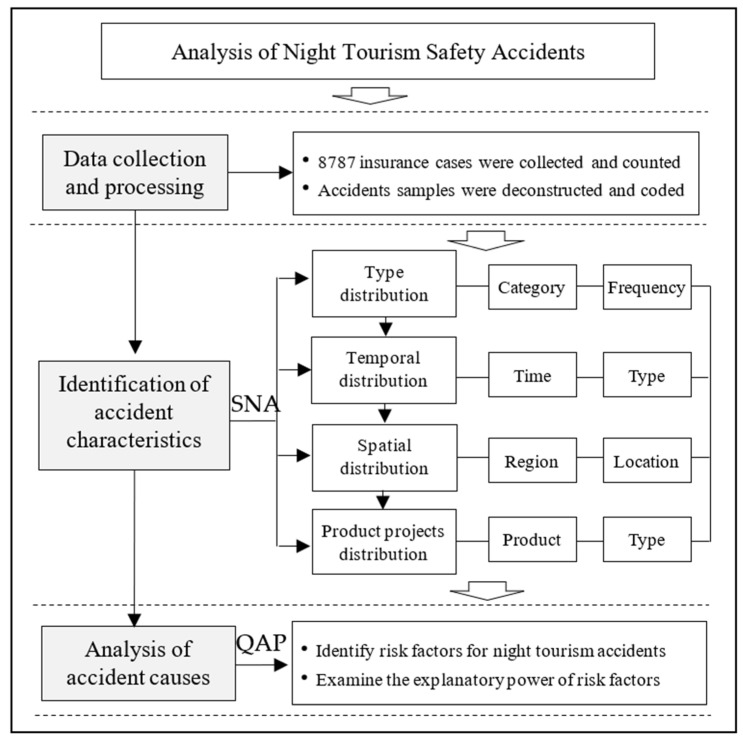
The analytical framework.

**Figure 2 ijerph-20-02584-f002:**
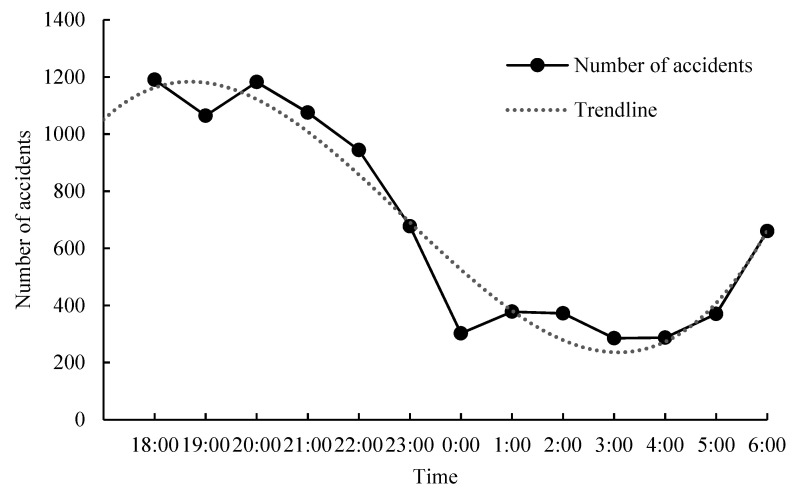
Temporal distribution of night tourism accidents.

**Figure 3 ijerph-20-02584-f003:**
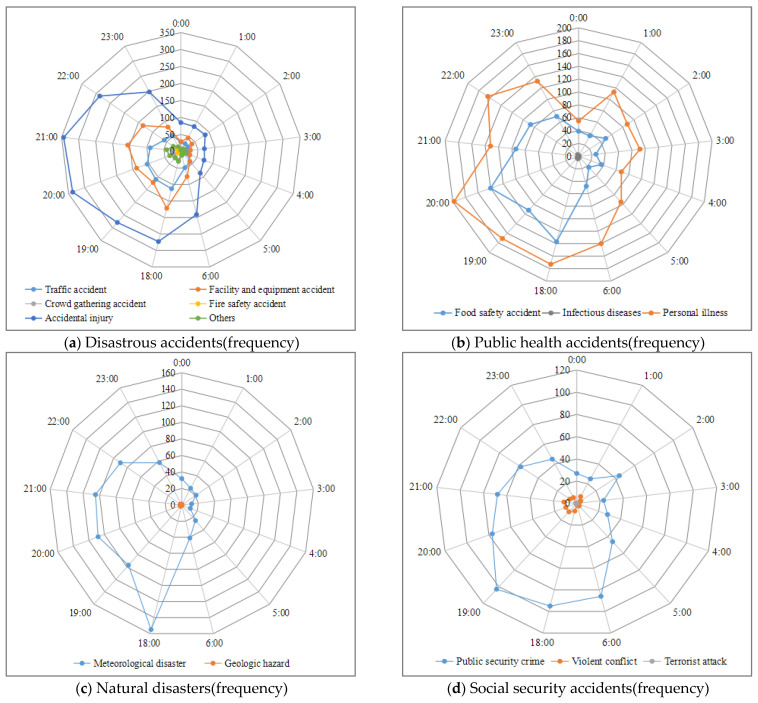
Temporal distribution of urban tourism accident types.

**Figure 4 ijerph-20-02584-f004:**
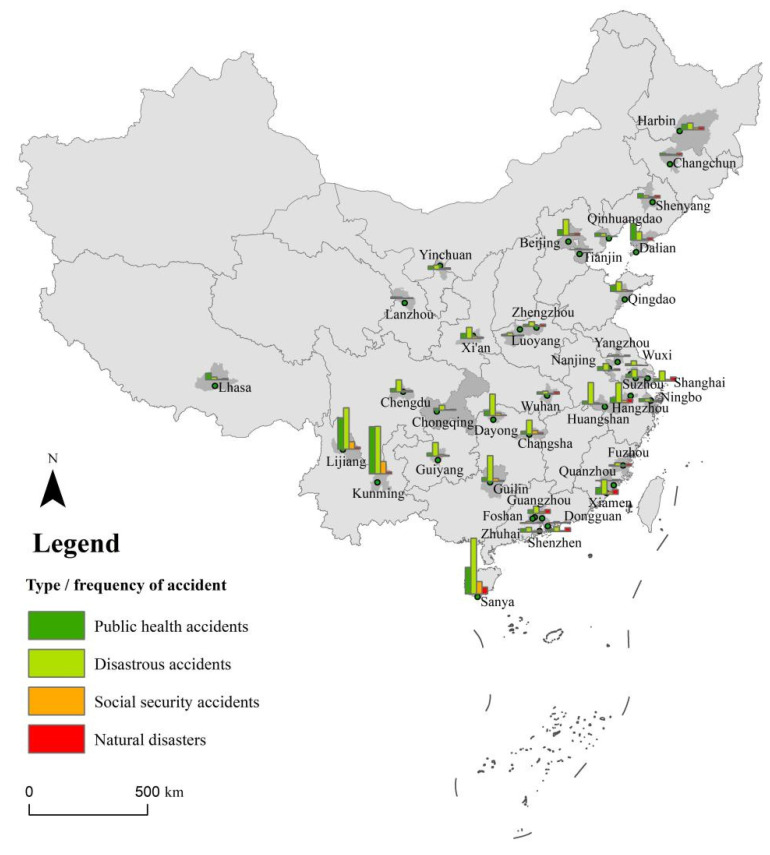
Frequency distribution of night tourism safety accidents in cities.

**Figure 5 ijerph-20-02584-f005:**
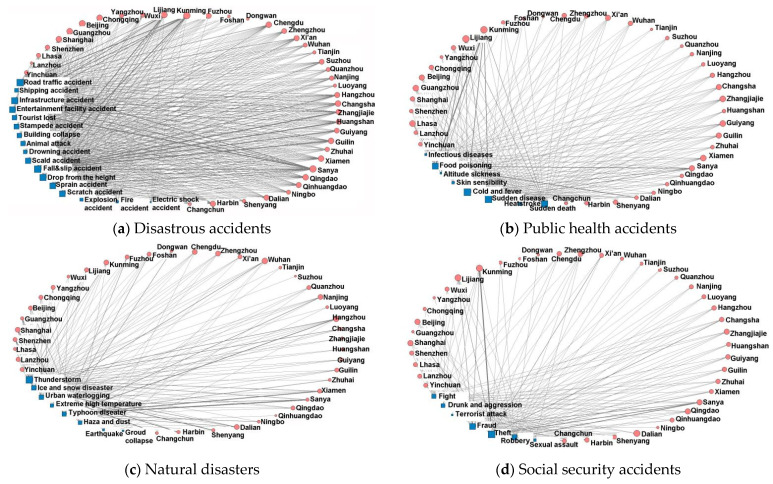
Two-mode structure network between subtype and cities.

**Figure 6 ijerph-20-02584-f006:**
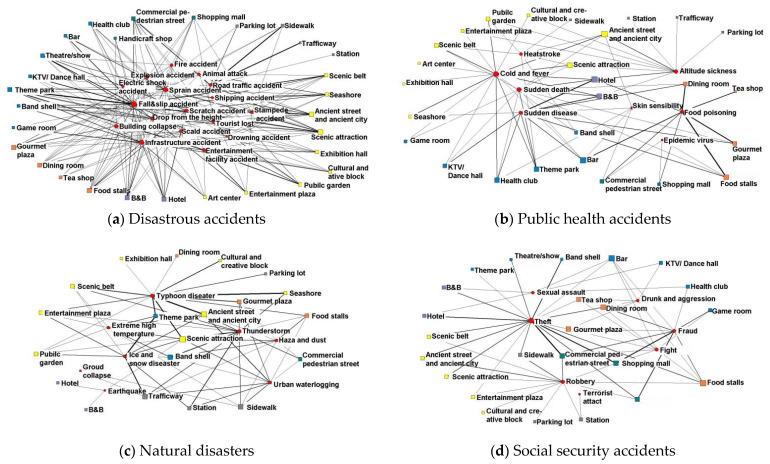
Correlation characteristics of site-accident types.

**Figure 7 ijerph-20-02584-f007:**
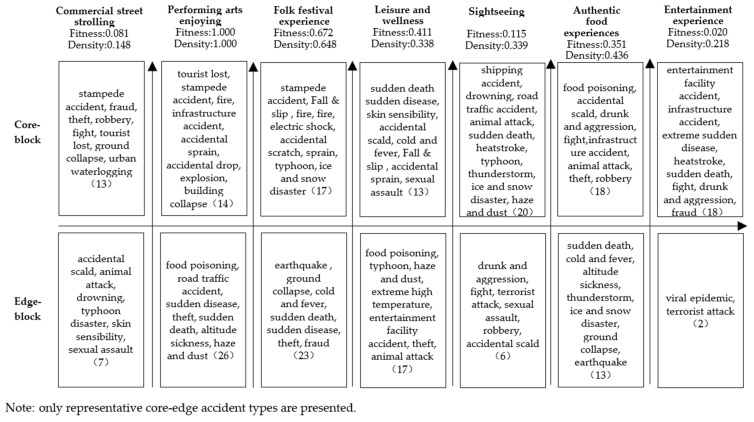
Core-edge structure of tourism accidents type in different product projects.

**Figure 8 ijerph-20-02584-f008:**
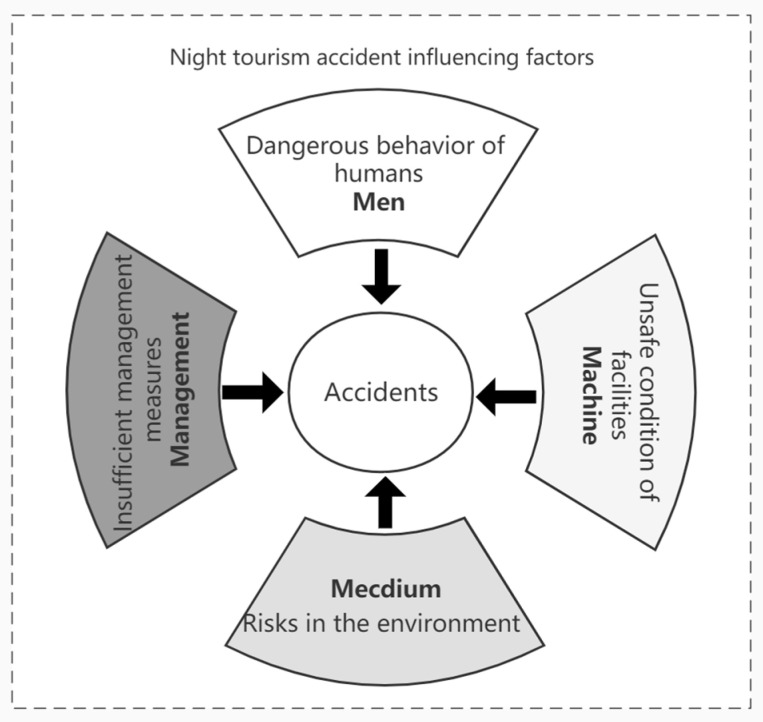
“4M” theoretical framework of accident causes.

**Table 1 ijerph-20-02584-t001:** Types of night tourism safety accidents in cities.

Main Type	Subtype	Basic Type	Count	Rate	Main Type	Subtype	Basic Type	Count	Rate
Disastrous accidents (53.19%)	Traffic accident	Road traffic accident	634	7.22%			Cold and fever	480	5.46%
		Shipping accident	102	1.16%			Sudden death	329	3.74%
	Facility and equipment accident	Entertainment facility accident	523	5.95%			Altitude sickness	170	1.93%
		Infrastructure accident	564	6.42%			Heatstroke	37	0.42%
	Crowd gathering accident	Stampede accident	111	1.26%	Natural disaster accidents (8.95%)	Meteorological disaster	Urban waterlogging	86	0.98%
		Lost tourist accident	59	0.67%			Haze and dust	53	0.60%
	Fire safety accident	Fire accident	16	0.18%			Extreme high temperature	49	0.56%
		Explosion accident	33	0.38%			Typhoon disaster	276	3.14%
		Electric shock accident	6	0.07%			Thunderstorm	167	1.90%
	Accidental injury	Fall and slip accidents	1368	15.57%			Ice and snow disaster	147	1.67%
		Scratch accident	328	3.73%		Geologic hazard	Ground collapse	3	0.03%
		Drop from the height	245	2.79%			Earthquake	6	0.07%
		Sprain accident	382	4.35%	Social security accidents (9.05%)	Public security crime	Theft	401	4.56%
		Scald accident	66	0.75%			Robbery	124	1.41%
	Others	Animal attack	101	1.15%			Fraud	193	2.20%
		Drowning accident	43	0.49%			Sexual assault	7	0.08%
		Building collapse	92	1.05%		Violent conflict	Drunk and aggression	31	0.35%
Public health accidents (28.81%)	Food safety accident	Food poisoning	897	10.21%			Fight	39	0.44%
	Infectious diseases	Epidemic virus	13	0.15%		Terrorism	Terrorist attack	1	0.01%
	Personal illness	Skin sensitivity	23	0.26%	Total	/		8787	100%
		Sudden disease	582	6.62%					

**Table 2 ijerph-20-02584-t002:** Network centrality measurement of city-accident types.

Accidents		Public Health Accidents		Natural Disasters		Social Security Accidents	
City	Node Centrality	City	Node Centrality	City	Node Centrality	City	Node Centrality
Sanya	1.000	Kunming	1.000	Hangzhou	0.750	Kunming	0.857
Kunming	1.000	Lijiang	1.000	Wuhan	0.750	Lijiang	0.857
Lijiang	0.941	Guangzhou	0.875	Chengdu	0.625	Dalian	0.857
Beijing	0.941	Guiyang	0.875	Dalian	0.625	Beijing	0.714
Shanghai	0.882	Lhasa	0.875	Kunming	0.625	Qingdao	0.714
Chengdu	0.824	Xiamen	0.875	Nanjing	0.625	Sanya	0.714
Changsha	0.824	Zhangjiajie	0.875	Shanghai	0.625	Shanghai	0.714
Guilin	0.824	Beijing	0.75	Zhengzhou	0.625	Zhangjiajie	0.714
Qingdao	0.824	Guilin	0.75	Beijing	0.500	Zhengzhou	0.714
Xiamen	0.824	Sanya	0.75	Foshan	0.500	Chengdu	0.571
Guangzhou	0.824	Xi’an	0.75	Fuzhou	0.500	Guiyang	0.571
Chongqing	0.824	Changsha	0.75	Guilin	0.500	Guilin	0.571
Dalian	0.765	Chengdu	0.625	Lanzhou	0.500	Harbin	0.571
Fuzhou	0.765	Hangzhou	0.625	Lijiang	0.500	Hangzhou	0.571
Harbin	0.765	Shanghai	0.625	Qingdao	0.500	Changsha	0.571
Hangzhou	0.765	Shenyang	0.625	Quanzhou	0.500	Zhuhai	0.571
Huangshan	0.765	Wuxi	0.625	Xiamen	0.500	Huangshan	0.429
Qinhuangdao	0.765	Wuhan	0.625	Shenzhen	0.500	Lhasa	0.429
Xi’an	0.765	Zhengzhou	0.625	Xi’an	0.500	Lanzhou	0.429
Zhangjiajie	0.765	Chongqing	0.625	Yangzhou	0.500	Luoyang	0.429

Notes: To simplify the data display content, only the degree center of main city fields is displayed.

**Table 3 ijerph-20-02584-t003:** Statistics of accident sites in night tourism of the cities.

Category	Site	Frequency	Rate
Catering place (15.90%)	Gourmet plaza	401	4.56%
	Dining room	343	3.90%
	Food stalls	533	6.07%
	Tea shop	121	1.38%
Accommodation (11.75%)	Hotel	644	7.33%
	B & B	388	4.42%
Traffic place (7.90%)	Parking area	100	1.14%
	Trafficway	266	3.03%
	Station	64	0.73%
	Sidewalk	264	3.00%
Places for sightseeing (24.34%)	Scenic belt	301	3.43%
	Ancient streets and towns	586	6.67%
	Scenic Attraction	565	6.43%
	Seashore	343	3.90%
	Art center	72	0.82%
	Cultural and creative block	129	1.47%
	Exhibition hall	143	1.63%
Shopping place (9.28%)	Commercial pedestrian street	451	5.13%
	Shopping mall	265	3.02%
	Handicraft shop	99	1.13%
Entertainment places (30.83%)	Entertainment Plaza	198	2.25%
	Public garden	296	3.37%
	Health club	185	2.11%
	Bar	361	4.11%
	KTV/Dance hall	215	2.45%
	Theme park	813	9.25%
	Game room	104	1.18%
	Theatre/show	261	2.97%
	Band shell	276	3.14%
Total	/	8787	100%

**Table 4 ijerph-20-02584-t004:** Network centrality measurement of site-accident types.

Disastrous Accidents		Public Health Accidents		Natural Disasters		Social Security Accidents	
Site	Node Centrality	Site	Node Centrality	Site	Node Centrality	Site	Node Centrality
Scenic Attraction	0.706	Hotel	0.625	Scenic Attraction	0.875	Bar	0.714
Ancient streets and town	0.647	Ancient street and town	0.625	Ancient streets and town	0.75	Food stalls	0.714
Gourmet plaza	0.588	Bar	0.625	Trafficway	0.625	Gourmet plaza	0.571
Commercial pedestrian street	0.588	Food stalls	0.5	Sidewalk	0.625	Commercial pedestrian street	0.571
Hotel	0.588	B & B	0.5	Band shell	0.625	Shopping mall	0.571
B & B	0.588	Scenic Attraction	0.5	Gourmet plaza	0.5	Dining room	0.571
Theme park	0.588	Health club	0.5	Food stalls	0.5	Tea shop	0.429
Food stalls	0.529	Theme park	0.5	Scenic belt	0.5	Handicraft shop	0.429
Dining room	0.471	Gourmet plaza	0.375	Entertainment Plaza	0.5	Hotel	0.286
Theatre/show	0.471	Dining room	0.375	Public garden	0.5	B & B	0.286
Scenic belt	0.412	Scenic belt	0.375	Theme park	0.5	Station	0.286
Seashore	0.412	Public garden	0.375	Station	0.375	Sidewalk	0.286
Shopping mall	0.412	Commercial pedestrian street	0.375	Commercial pedestrian street	0.375	Scenic belt	0.286
Health club	0.412	KTV/Dance hall	0.375	Seashore	0.25	Ancient streets and town	0.286
Exhibition hall	0.353	Seashore	0.25	Dining room	0.125	Scenic Attraction	0.286
Public garden	0.353	Cultural and creative block	0.25	Hotel	0.125	Entertainment Plaza	0.286
Art center	0.294	Entertainment Plaza	0.25	B & B	0.125	Health club	0.286
Cultural and creative block	0.294	Shopping mall	0.25	Parking area	0.125	KTV/Dance hall	0.286
KTV/Dance hall	0.294	Game room	0.25	Cultural and creative block	0.125	Game room	0.286
Band shell	0.294	Band shell	0.25	Exhibition hall	0.125	Parking area	0.143

Notes: To simplify the data display content, only the degree center of the main site is displayed.

**Table 5 ijerph-20-02584-t005:** Statistics of risk factors of nighttime tourism safety accidents.

Category	Cause	Frequency	Category	Cause	Frequency
Personnel risk factors (33.97%)	Tourists’ awareness	10.83%	Environmental risk factors (24.16%)	Atmospheric environment	8.85%
	Tourists’ physical condition	16.25%		Geological environment	2.00%
	Employees’ safety service	6.09%		Road environment	7.22%
	Conflict between local residents and tourists	0.80%		Tourism environment	6.09%
Management risk factors (24.29%)	Market supervision and management	6.89%	Facility and equipment risk factors (17.58%)	Lighting facility	2.79%
	Social security management	6.07%		Extinguishing facility	1.38%
	Public health management	9.40%		Entertainment facility	5.95%
	Crowd aggregation control	1.93%		Engineering facility	7.47%

**Table 6 ijerph-20-02584-t006:** Results of the QAP regression analysis.

Independent Variable Co-Occurrence	18:00–6:00	18:00–20:00	21:00–23:00	0:00–2:00	3:00–6:00
Personnel risk	0.227 ***	0.261 ***	0.194 ***	0.477 ***	0.453 ***
Facility and equipment risk	0.344 ***	0.260 ***	0.380 ***	0.065	0.215 ***
Environmental risk	0.165 ***	0.269 ***	0.247 ***	0.139 ***	0.237 ***
Management risk	0.315 ***	0.297 ***	0.255 ***	0.351 ***	0.063
R^2^	0.976	0.977	0.974	0.828	0.768
Adjusted R^2^	0.976	0.977	0.974	0.827	0.768

Note: *** *p* < 0.001.

## Data Availability

The data used to support the findings of this study are available from the corresponding author upon request.

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
