# Peer review of "Analysis of the Characteristics and Causes of Night Tourism Accidents in China Based on SNA and QAP Methods"

_ijerph, 2023, doi:10.3390/ijerph20032584_

Round 1
Reviewer 1 Report
Please include the following comments:
1. English proofread is required for the manuscript.
2. Abstract- Mention abbreviation QAP full form. (Quadratic Assignment Procedure)
What is list of managerial implications?
What is the theoretical implication for this study?
Any behavioral pattern of the night commuters?
How were the study area cities selected?
What do you mean by "third, in natural disaster accidents"?
Please check format for the references.
Good Luck
Author Response
If you would like to see a clearer response, please see the attachment
At the outset, we would like to thank the editor and all reviewers for allowing us to improve our paper and for their valuable comments. We have made many revisions based on the reviewers’ feedback and believe that the quality of our paper has increased as a result. Changes have been highlighted in yellow in the attached manuscript.
1. English proofread is required for the manuscript.
Thank you for your suggestion.
We are very sorry for the language issues in the original manuscript and for the inconvenience caused to the reviewers in the process. The grammar, spelling and sentence structure of the article were rechecked and improved with assistance from a native English speaker with appropriate research background. We hope that the revised proofread manuscript is now acceptable to the publisher.
2. Abstract-Mention abbreviation QAP full form. (Quadratic Assignment Procedure)
Thank you for your comment. We have now written in the full form instead of the acronym form in the Abstract section. The revisions were as follows:
“Abstract:......This research takes a novel methodological approach, by using 8,787 cases of tourist safety accidents in typical night tourism cities in China, and apply social network analysis and Quadratic Assignment Procedure (QAP) regression analysis to explore the multidimensional structural characteristics and risk-causing factors of night tourism accidents......”
The other parts of the manuscript were also updated.
3. What is list of managerial implications?
Thank you for your comments about the “list of managerial implications”.
We apologize for the lack of clarity in the “list of managerial implications” in the Abstract section. The original statement was: we have put forward some practical suggestions for government to control night tourism safety accidents of cities in the Managerial implications section. However, we could not list these specific suggestions all in the Abstract section due to space limitations. Therefore, we used the sentence “A list of managerial implications is suggested for further development” to describe what we have done. In order to make the readers grasp the information more clearly, we have made the following revision to this section(Page 1, line 22-23):
“Abstract: ......management has a high explanatory power on the accident correlation level in the co-occurrence network of urban night tourism safety accidents, and the influence effects of risk factors are heterogeneous at different time points. Our results provide some valuable implications for optimizing urban night tourism safety governance.”
4. What is the theoretical implication for this study?
Thank you for your question.
We believe that the theoretical implications of our paper are mainly reflected in two aspects. We have supplemented theoretical implications in the revised manuscript in response to your concerns. The revisions were as follows(Page 18,Line 525-546):
“The theoretical implications of our study are mainly reflected in two aspects. First, there is no doubt that safety plays a critical role in the development of night tourism, but the safety issue of night tourism has received little attention. The limited studies on night tourism safety accidents have been mostly restricted to a single city or a single night accident type such as crime accidents, food poisoning accidents and traffic accidents [13,16,17;39; 60]. However, few studies have been able to systematically extract and summarize the common characteristics of tourism safety accidents in different cities. To the best of our knowledge, this is the first study based on large-scale insurance data to comprehensively reveal the multidimensional structural characteristics of night tourism accidents, including spatial-temporal distribution characteristics, type characteristics and product classification characteristics, which provides the basis for understanding the common sense of the night tourism accidents. Second, the occurrence of night tourism safety accidents is influenced by the complex and dynamic risk factors. Some studies have explored single influencing factors of night tourism accidents, such as abusive consumption of alcohol, overcrowding, and street lighting conditions [39,40,61]. However, there remain gaps in the systematic analysis of which factors affect night tourism safety accident based on specific theoretical framework. As a key contribution to the advancement of theory, this studies identified the diverse risk factors leading to night tourism accidents rooted in the 4M Theory. Although the 4M Theory has been widely recognized and applied in the fields of safety science re-search [62, 63], few studies have used it to explain the phenomenon of night tourism safety accidents. Our research further expands the theoretical application scenario of the 4M Theory.”
5. Any behavioral pattern of the night commuters?
Thank you very much for pointing out this potential contribution which we have now elaborated in the revised paper.
Commuters and tourists are two important components of non-residential urban population. However, previous studies have found significant differences between tourists and commuters. Commuters are those who maintain domicile outside or in the suburbs of the city, but working or studying in the city. Tourists are people traveling to and staying in places outside their usual environment for leisure, business and other purposes, for a limited amount of time (Mamei & Colonna, 2018). Some studies conceptualize commuters as individuals who travel between the city and their home in the greater metropolitan area on a regular, consistent basis, and conceptualize tourists as intermittent visitors who live outside of the metropolitan area, if not another city entirely (Tucker et al., 2021). In other words, the starting point, ending point and scope of the night commuter's movement are regular, and the surrounding environment is very familiar and resort to mundane routines. In contrast, tourists are often in a completely unfamiliar environment, and their traveling route has greater mobility (Löfgren, 2015;Mamei & Colonna, 2018; Wu et al., 2020), which means that the behavioral pattern of the tourists often have higher risks(Tucker et al., 2021). Therefore, we choose tourists as our research objects or unit of analysis. In previous studies, it is difficult to distinguish between tourists’ data and commuters’ data (Mamei & Colonna, 2018), but our data from the insurance company is specifically targeted at tourist groups, which is the strength of this research.
In response to the questions raised by experts, we have added corresponding contents in the Literature review section(Page 3, Line111-113):
“Compared with commuters who regularly and consistently travel between cities and places of residence, tourists are often in a completely unfamiliar environment, and their traveling route has greater mobility (Mamei & Colonna, 2018; Wu et al., 2020), which means that the behavioral pattern of the tourists often have higher risks (Löfgren, 2015; Tucker et al., 2021).”
References:
- Löfgren, O. (2015). Modes and moods of mobility: Tourists and commuters. Culture Unbound, 7(2), 175-195.
- Mamei, M., & Colonna, M. (2018). Analysis of tourist classification from cellular network data. Journal of Location Based Services, 12(1), 19-39.
- Wu, Y., Wang, L., Fan, L., Yang, M., Zhang, Y., & Feng, Y. (2020). Comparison of the spatiotemporal mobility patterns among typical subgroups of the actual population with mobile phone data: A case study of Beijing. Cities, 100, 102670.
- Tucker, R., O’Brien, D. T., Ciomek, A., Castro, E., Wang, Q., & Phillips, N. E. (2021). Who ‘tweets’ where and when, and how does it help understand crime rates at places? Measuring the presence of tourists and commuters in ambient populations. Journal of Quantitative Criminology, 37(2), 333-359.
6. How were the study area cities selected?
Thank you for pointing this out.
In order to ensure the representativeness of the study area, our cities have been carefully considered. The 40 sample cities selected for the study were based on two national research reports: China Night-time Economy Development Report (2020) and China Urban Leisure and Tourism Competitiveness Report (2020).
First, the China Tourism Academy announced the "Top 20 Cities of China's Night Economy in 2020" in the China Night-time Economy Development Report (2020. We have included these 20 cities into the research sample. Second, the China Urban Leisure and Tourism Competitiveness Report (2020) also listed the 20 typical and most popular night tourism and leisure cities, which we have added as a supplement to the first report. We believe that the development of a large number of night tourism activities is an essential prerequisite for the occurrence of night tourism safety accidents, which is also supported by the insurance data. Therefore, we take the well-developed night tourism destinations as the main samples to study night tourism safety accidents.
To further address the reviewer’s concerns, we have added a more detailed interpretation regarding how we have selected the study area in the Data source and processing section. The revisions were as follows(page 4,line162-166):
“The 40 sample cities selected for this study comprised of (a) the top 20 typical nighttime economic cities in China selected from the China Night-time Economy Development Report (2020) issued by the China Tourism Academy, and (b) 20 typical leisure tourism cities selected from the China Urban Leisure and Tourism Competitiveness Report (2020).”
7. What do you mean by "third, in natural disaster accidents"?
Thank you for your question.
In the original manuscript, “Third, in natural disaster accidents” actually means “Third, in the relationship structure network between natural disasters and cities (Figure 4c.).” The rationale is to show how tourism-related natural disaster accidents are distributed in each city.
To clarify this, we have revised the section as follows: The revised sentence was as follows:
“Third, the structure network between natural disaster accidents and cities (Figure 4c.) shows that Hangzhou, Wuhan, Chengdu, Dalian and Kunming have a large degree of point centrality, but the types of natural disaster accidents associated with such cities are complex. In terms of node connection strength, rainstorm, lightning and typhoon disasters are highly correlated with coastal tourist cities such as Sanya, Xiamen and Guangzhou. Urban waterlogging is highly correlated with mega cities such as Beijing and Shanghai, as well as historical and cultural tourist cities such as Wuhan, Changsha and Hangzhou. Ice and snow disasters are highly correlated with Harbin, Shenyang, Changchun, which are northern cities in China dominated by ice and snow tourism. Extreme high temperatures are highly correlated with cities such as Sanya, Xiamen, Chongqing and Changsha. Haze and dust are strongly correlated with Beijing, Zhengzhou and Lanzhou.”
8. Please check format for the references.
Thanks for your suggestion.
In our resubmitted manuscript, we have revised the format of all references according to the requirements of the journal.

Reviewer 2 Report
The subject, no doubt, is of interest for many reasons. Yet the evidence put forward in the paper contains a minimum of surprises, of noteworthy discovery. And you need a proper, focused conclusion, dealing with specific measures found helpful in your research addressing remedies for night time tourism hazards.
Author Response
If you would like to see a clearer response, please see the attachment
At the outset, we would like to thank the editor and all reviewers for allowing us to improve our paper and for their valuable comments. We have made many revisions based on the reviewers’ feedback and believe that the quality of our paper has increased as a result. Changes have been highlighted in yellow in the attached manuscript.
1. The subject, no doubt, is of interest for many reasons. Yet the evidence put forward in the paper contains a minimum of surprises, of noteworthy discovery. And you need a proper, focused conclusion, dealing with specific measures found helpful in your research addressing remedies for night time tourism hazards.
Thank you for your valuable feedback.
We believe that our research contributions are mainly reflected in the following four aspects:
First, notwithstanding the critical role that safety plays in night tourism, safety issues during night time travel have received scant attention. The limited studies on night tourism safety accidents have been mostly restricted to a single city or a single night accident type such as crime accidents, food poisoning accidents, traffic accidents, etc. (Bishop & Robinson, 1999; Hsieh & Chang, 2006; Tutenges, 2009; Liu et al., 2021; Wang et al., 2022). However, few studies have been able to systematically extract and summarize the common characteristics of tourism safety accidents in different cities. To the best of our knowledge, this is the first study based on large-scale insurance data to comprehensively reveal the multidimensional structural characteristics of night tourism accidents, including spatial-temporal distribution characteristics, type characteristics and product classification characteristics, which provides the basis for understanding the common sense of the night tourism accidents.
Second, the occurrence of night tourism safety accidents is influenced by the complex and dynamic risk factors. Some studies have explored single influencing factors of the night tourism accidents, such as abusive consumption of alcohol, overcrowding, and street lighting conditions (Tutenges, 2009; Song et al., 2020; Calafat et al., 2011). However, there remain gaps in the systematic analysis of which factors affect night tourism safety accident based on specific theoretical framework. As a key contribution to the advancement of theory, this study identified the diverse risk factors leading to night tourism accidents rooted in 4M Theory. Although the 4M Theory has been widely recognized and applied in the fields of safety science research (Mao and Xu 2011; Song and Xie 2014; Jamot &Park 2019 ;), few studies have used it to explain the formation of night tourism safety accidents. Our research further expands the theoretical application scenario of 4M Theory.
Third, most existing studies are qualitative and descriptive in nature and only focused on single risk types of night tourism (Tutenges, 2009; Liu et al., 2021; Wang et al., 2022). The importance and originality of this study are that it takes a novel methodological approach, by using 8,787 cases of tourist safety accidents in typical night tourism cities in China, and apply social network analysis and Quadratic Assignment Procedure (QAP) regression analysis to explore the multidimensional structural characteristics and risk-causing factors of night tourism accidents. It is a novel contribution that applies the quantitative approach to explore the night tourism safety accidents.
Fourth, most of the previous studies on night safety accidents in cities were about urban residents or commuters(Tucker et al., 2021). Due to the lack of data, few studies carried out systematic research on tourist groups. In fact, compared with the first two groups, tourists are often in a completely unfamiliar environment, and their traveling route has greater mobility (Mamei & Colonna, 2018; Wu et al., 2020), which means that the behavioral pattern of the tourists often have higher risks (Löfgren, 2015; Tucker et al., 2021). And our data from the insurance company is specifically targeted at tourist groups, which is the strength of this research in terms of research population.
In addition, the conclusions of this study can be used to develop targetted interventions aimed at avoiding the night tourism accidents. Some of the specific suggestions we put forward are shown in the table below:
Check list of the conclusions and countermeasures
|
Conclusions |
Countermeasures |
|
| 1 |
The types of night tourism accidents in the cities show complex and diversified classification characteristics...... |
There is a need to strengthen the cooperation between government departments and enhance the ability of collaborative governance for multiple accidents...... |
| 2 |
Aggregation and heterogeneity exist in the temporal and spatial distribution of night tourism safety accidents in the cities...... |
There should be a focus on offering a differentiated management and control countermeasure, based on the unbalanced spatial-temporal distribution of accidents...... |
| 3 |
Structural differences exist in accident types in night tourism product projects. Among the 7 categories of night tourism products, the accidents in 4 categories of products have obvious core-edge structure...... |
Plans must be implemented to strictly supervise the market of product projects and strengthen the safety control of high-incidence accident types...... |
| 4 |
The risk inducing factors of night tourism accidents are divided into four structural aspects: personnel risk, facilities and equipment risk, environmental risk and management risk...... |
More importantly, there is a need to establish a systematic and normalized safety governance system based on multiple risk factors. First, it is necessary to...... |
| 5 |
In addition, the explanatory power of risk factors on night tourism accident are different at different time periods. |
In addition, it is necessary to focus on the intervention and guidance for the leading risk factors of night tourism at different time periods. |
We have revised the manuscript to address your concerns, and hope our contribution can be clearer. Please see page 3, lines 111–113, page 4, lines 145–152, and page 18, lines 524–546 in the revised manuscript.
If there are any other modifications we could make, we would like very much to address them and we really appreciate your help. Thank you very much!

Reviewer 3 Report
Please be consistent about the terminology usage. (night tourism, night-time tourism, urban night tourism)
There can be found quite a lot of minor grammatical errors. Please check.
Figure 3 inset map has broken bars, and the exact region in the inset mat is not clear. Please correct them.
Text in figure 4 are not visible. Please consider enhancing the visibility of the diagram.
Please also take care of the reference style.
Author Response
If you would like to see a clearer response, please see the attachment
At the outset, we would like to thank the editor and all reviewers for allowing us to improve our paper and for their valuable comments. We have made many revisions based on the reviewers’ feedback and believe that the quality of our paper has increased as a result. Changes have been highlighted in yellow in the attached manuscript.
1. Please be consistent about the terminology usage. (night tourism, night-time tourism, urban night tourism) There can be found quite a lot of minor grammatical errors. Please check.
Thank you bringing up the consistency of terminology usage.
We do apologize for this oversight. Referring to the common expression in the existing literature (Chen et al., 2020; Li et al., 2022), the terminology has been uniformly modified to “night tourism” in our revised manuscript. This is also most relevant to our study.
In addition, the grammatical errors of this article were rechecked and corrected with assistance from a native English speaker with the appropriate research background. We sincerely hope the revised manuscript is acceptable in terms of English structure.
2. Figure 3 inset map has broken bars, and the exact region in the inset mat is not clear. Please correct them.
Thank you for pointing this out.
According to the reviewer’ suggestions and the presentation of China's maps in similar studies in the International Journal of Environmental Research and Public Health, we have re-adjusted the map to make it clearer. It should be noted that the broken bars on the map is the national boundary of China. We decided to keep it after referring to two other papers published in this journal (Zhuang et al., 2022; Wang & Zhang, 2022).
Reference sources for maps:
1.Zhuang, R., Mi, K., Zhi, M., & Zhang, C. (2022). Digital Finance and Green Development: Characteristics, Mechanisms, and Empirical Evidences. International Journal of Environmental Research and Public Health, 19(24), 16940.
2.Wang, J., & Zhang, G. (2022). Dynamic Evolution, Regional Differences, and Spatial Spillover Effects of Urban Ecological Welfare Performance in China from the Perspective of Ecological Value. International Journal of Environmental Research and Public Health, 19(23), 16271.
3. Text in figure 4 are not visible. Please consider enhancing the visibility of the diagram.
Thanks for pointing this out.
We apologize for the inconvenience caused by the unclear diagrams. The text in figure 4 has now been revised. In addition, we have also checked and modified other diagrams for consistency.
4. Please also take care of the reference style.
Thanks for your suggestion.
In the resubmitted manuscript, we have revised the format of all references according to the requirements of the journal.

Round 2
Reviewer 2 Report
I judge the revised manuscript, despite the originality of its focus, to be of little scholarly significance, not enough to merit publication. My original judgments expressed to you remain unaffected by the minor revisions in grammar, style, and repetitions.
Author Response
We recommend that you check the complete reply in the attachment.
Reviewer: I judge the revised manuscript, despite the originality of its focus, to be of little scholarly significance, not enough to merit publication. My original judgments expressed to you remain unaffected by the minor revisions in grammar, style, and repetitions.
Thank you for your comment!We hope you read all of our responses before making a decision.
While we appreciate the reviewer’s feedback, we respectfully disagree with your view that little scholarly of our article. First, in terms of research types, it is generally believed that the scientific research can be grouped into three types: exploratory, descriptive and explanatory(Bhattacherjee, 2012; Liu et al., 2020)[1][2]. Obviously, our research is a combination of descriptive research and explanatory research. On the one hand, we describe the basic structural characteristics of night tourism accidents, including spatial-temporal distribution characteristics, type characteristics and product classification characteristics. On the other hand, we seek to explain why the night tourism safety accidents are happening based on the observed phenomenon and determine the influential effect of each variable in different time periods.
Second, in terms of research philosophy, induction and deduction are two aspects of scientific research(Lawson,2005; Woiceshyn et al., 2018)[3][4]. This study systematically extract and summarize the common characteristics of tourism safety accidents in different cities based on the inductive logic, which is a process from specific observations to broader generalizations. In addition, according to the 4M theory, we have confirmed the impact of four risk factors on night tourism safety accidents, reflecting the application of deductive logic in this study.
Third, in terms of research contents, the analysis of accident characteristics and causes based on the statistical data is widely recognized as an important part in the research field of safety science. In many studies of safety accidents such as construction accidents[5], traffic accidents[6], occupational accidents[7] and coal mine accidents[8], the main research objective is to analyze the characteristics and causes of accidents, because the findings form these researches can provide a basis for the development of accident prevention measures for the government and enterprise. In addition, the research contents on the characteristics and causes of accidents are also very common in international representative journals such as Safety Science, Accident Analysis and Prevention, Process Safety and Environmental Protection and Journal of Safety Research[5-8]. Therefore, our research contents and paradigm are worthy of recognition in the field of safety science research.
Fourth, in terms of research methodology, the Social Network Analysis (SNA) has been widely applied to identify the complex accident characteristics, and Quadratic Assignment Procedure (QAP) is especially useful for analyzing risk factors of the accidents[9][10][11][12]. The method of Social Network Analysis (SNA) and Quadratic Assignment Procedure (QAP) are mature techniques for accident analysis. Therefore, we believe that the results obtained by these methods are reliable.
From the above discussion, we consider our article to be of academic merit. In fact, the subject of our article fits well with the International Journal of Environmental Research and Public Health, which we found through statistics that the International Journal of Environmental Research and Public Health has published a large number of studies in recent years on the characteristics and causes of accidents in different fields (shown in Table 1) . However, few studies have been able to summarise the characteristics of night tourism safety incidents and explore their causes based on representative data, although night tourism safety is crucial to the development of night tourism industry. Our teams and institutions have accumulated more than 20 years of professional experience in the field of tourism safety research, and have also obtained the certification of the World Tourism Organization. As a researcher, as well as a reviewer, I do believe that our research is very valuable in promoting safety development of the night tourism industry. In particular, this studies conducted a systematic investigation of night tourism accident based on the nation-wide insurance data, and some practical implications for optimizing night tourism safety governance of the cities were provided. We hope that the safety and security of more urban tourists can benefit from our articles.
Table 1. Studies related to accident characteristics and causes
|
Title |
Author |
Journal |
Area |
Description |
|
Characteristics and Causes of Particularly Major Road Traffic Accidents Involving Commercial Vehicles in China |
Yan,et al.(2021) |
International Joumal of Environmental Research and Public Health |
Traffic Accidents |
This study investigated 11 particularly major accidents involving commercial vehicles in China, and performed analysis on accident characteristics regarding the time, location, types of vehicles, and accident causation at different levels based on the 24Model. |
|
Characteristics and Statistical Analysis of Large and above Hazardous Chemical Accidents in China from 2000 to 2020 |
Yang,et al.(2022)
|
International Joumal of Environmental Research and Public Health |
Hazardous Chemical Accidents |
A general description of the characteristics of larger and above accidents based on 195 large and above accidents of hazardous chemicals in China, and then the system risk of the hazardous chemical industry was calculated and evaluated. |
|
Characteristics, Cause, and Severity Analysis for Hazmat Transportation Risk Management
|
Zhou,et al.(2020) |
International Joumal of Environmental Research and Public Health |
Hazmat Transportation Accidents |
This study firstly analyzed spatial–temporal trends to understand the major characteristics of hazmat transportation accidents. Secondly , it presented a quantitative description of the relation among the hazmat properties, accident characteristics, and the consequences of the accidents using the decision tree approach. |
|
Types and Characteristics of Fatal Accidents Caused by Multiple Processes in a Workplace: Based on Actual Cases in South Korea |
Kang,et al.(2022)
|
International Joumal of Environmental Research and Public Health |
Industrial fatal accidents |
This study classified accidental types and their characteristics based on actual cases, in which potential risks exist at multiple processes in a workplace. |
|
Analysis of the Characteristics of Fatal Accidents in the Construction Industry in China Based on Statistical Data |
Xu,et al.(2021) |
International Joumal of Environmental Research and Public Health |
Construction accidents |
This study aimed to reveal the characteristics of 6005 fatal accidents in the construction industry in China from 2010 to 2019. The important features of these fatal accidents, such as the type, time of occurrence, site location, severity, and geographical region of the accident, were carefully analyzed. |
|
Risk Factors Affecting T raffic Accidents at Urban Weaving Sections: Evidence from China |
Mao, et al.(2019) |
International Joumal of Environmental Research and Public Health |
Traffic Accidents |
We employed the multinomial logistic regression approach to identify the correlation between six categories of risk factors and four types of traffic accidents based on 768 accident samples of an observed weaving section from 2016 to 2018 |
|
Characteristics and Causes of Construction Accidents in a Large-Scale Development Project |
Chan, et al.(2022) |
Sustainability |
Construction Accidents |
Through the detailed inspection of 8 construction accidents, the modified loss causal relationship model is applied to analyze the situation variables, event sequences and accident causes. |
|
Pietilä J, Räsänen T, Reiman A, et al. Characteristics and determinants of recurrent occupational accidents[J]. Safety science, 2018, 108: 269-277. |
JuliaPietilä, et al.(2018) |
Safety Science |
Occupational accidents |
They used a data set of a Finnish insurance company to explore the characteristics and determination of recurrent occupational accidents. |
|
Focusing on the patterns and characteristics of extraordinarily severe gas explosion accidents in Chinese coal mines |
zhang,et al(2018) |
Process Safety and Environmental Protection |
Gas explosion accidents |
They used 126 extraordinarily severe gas explosion accidents to assess statistical characteristics about accident-related factors, such as gas accumulation, ignition sources, operating locations, accident time, coal mine regions and coal mine ownership. |
Last but not least, in order to highlight the academic nature of this research, we refer to the title of the existing safety accident research in International Joumal of Environmental Research and Public Health, and change the title of this paper to “Analysis of the Characteristics and Causes of the Night Tourism Accidents in China Based on SNA and QAP Methods”. In addition, compared with the original manuscript, we have made a great revisions based on the reviewers’ feedback to reflect the academic value of this paper. The specific revisions are as follows:
(1.Introduction: page 2, lines 73–90)
“To fill this gap, this research takes a novel methodological approach, by using 8,787 cases of tourist safety accidents in typical night tourism cities in China, and apply social network analysis(SNA) and Quadratic Assignment Procedure (QAP) regression analysis to explore the multidimensional structural characteristics and risk-causing factors of night tourism accidents(Figure 1.). This research makes the following contributions. Firstly, we establish the relationship network among related elements of accidents, such as the type, time of occurrence, site location, and geographical region, which reveals the common characteristics of night tourism accidents in different cities. Secondly, based on 4M theory framework, we precisely and comprehensively identify 16 sub-risk factors and 4 major risk factors affecting night tourism accidents. And we also examine the effects of personnel, facilities, environment and management on tourism safety accidents at different time periods with the help of the QAP in SNA. The key purpose of this paper is to provide a theoretical reference for enhancing night tourism safety governance in cities. The remainder of this paper proceeds as follows. The second section reviews the relevant literature. The third section describes the data processing and method application. The fourth section presents the analytical process and discussion. The fifth section presents conclusions and implications.”
(2.4 Gaps in the literature: page 4, lines 166–185)
“Night tourism accidents are not rare in China's urban tourism industry and have resulted in a large number of fatalities and substantial property damage. However, To date, far too little attention has been paid to night tourism accidents. First, most existing studies are qualitative and descriptive in nature and only focused on single risk types of night tourism. What is inherently deficient is an overarching approach in summarizing the structural characteristics of the night tourism safety accidents and comparing different cities. The SNA method is based on the co-occurrence relationship of nodes as the examination element, which is suitable for revealing the overall characteristics of night tourism accidents via relational data. This paper serves to close this gap by applying a social network analysis approach with a unique classification across typical night tourism cities in China as case sites to provide a theoretical reference for optimizing night tourism safety governance in cities. Second, there are few researches on the influencing factors of night tourism accidents based on large-scale data and lack of relevant theoretical support. It is necessary to propose a theoretical framework to establish the correlation between risk factors and night tourism accidents, and compare the impacts of various risk factors on the accidents with the help of the QAP in SNA. Once we have a better understanding of the causes of night tourism accidents, corresponding measures can be adopted to improve the safety of tourists.”
(6.2. Theoretical implications: page 20, lines 562–583)
“The theoretical implications of our study are mainly reflected in two aspects. First, there is no doubt that safety plays a critical role in the development of night tourism, but the safety issue of night tourism has received little attention. The limited studies on night tourism safety accidents have been mostly restricted to a single city or a single night accident type such as crime accidents, food poisoning accidents and traffic accidents [13,16,17; 39,60]. However, few studies have been able to systematically extract and summarize the common characteristics of tourism safety accidents in different cities. To the best of our knowledge, this is the first study based on large-scale insurance data to comprehensively reveal the multidimensional structural characteristics of night tourism accidents, including spatial-temporal distribution characteristics, type characteristics and product classification characteristics, which provides the basis for understanding the common sense of the night tourism accidents. Second, the occurrence of night tourism safety accidents is influenced by the complex and dynamic risk factors. Some studies have explored single influencing factors of night tourism accidents, such as abusive consumption of alcohol, overcrowding, and street lighting conditions [39, 40, 61]. However, there remain gaps in the systematic analysis of which factors affect night tourism safety accident based on specific theoretical framework. As a key contribution to the advancement of theory, this studies identified the diverse risk factors leading to night tourism accidents rooted in the 4M Theory. Although the 4M Theory has been widely recognized and applied in the fields of safety science research [62, 63, 65], few studies have used it to explain the phenomenon of night tourism safety accidents. Our research further expands the theoretical application scenario of the 4M Theory.”
REFERENCES
[1] Bhattacherjee A. Social science research: Principles, methods, and practices[J]. 2012.
[2] Liu W, Liang Y, Wei S, et al. The organizational collaboration framework of smart logistics ecological chain: a multi-case study in China[J]. Industrial Management & Data Systems, 2020.
[3] Lawson A E. What is the role of induction and deduction in reasoning and scientific inquiry?[J]. Journal of Research in Science Teaching, 2005, 42(6): 716-740.
[4] Woiceshyn J, Daellenbach U. Evaluating inductive vs deductive research in management studies: Implications for authors, editors, and reviewers[J]. Qualitative research in organizations and management: An International Journal, 2018, 13(2): 183-195.
[5] Chan A P C, Yang Y, Choi T N Y, et al. Characteristics and Causes of Construction Accidents in a Large-Scale Development Project[J]. Sustainability, 2022, 14(8): 4449.
[6] Yan M, Chen W, Wang J, et al. Characteristics and causes of particularly major road traffic accidents involving commercial vehicles in China[J]. International journal of environmental research and public health, 2021, 18(8): 3878.
[7] Pietilä J, Räsänen T, Reiman A, et al. Characteristics and determinants of recurrent occupational accidents[J]. Safety science, 2018, 108: 269-277.
[8] Zhang J, Cliff D, Xu K, et al. Focusing on the patterns and characteristics of extraordinarily severe gas explosion accidents in Chinese coal mines[J]. Process Safety and Environmental Protection, 2018, 117: 390-398.
[9] Eteifa, S.O. ; El-adaway, I.H. Using social network analysis to model the interaction between root causes of fatalities in the construction industry. Journal of Management in Engineering 2018, 34, 04017045.
[10] Li, Z.M.; Wang, R. Research on characteristics of expressway truck accidents in different regions. China Safety Science Journal 2020, 30, 121-127.
[11] Zhang, K.; Xie, C.W. Research on temporal and spatial distribution and causes of water-related tourism safety accidents in China. Journal of Safety Science and Technology 2020, 8, 167-172.
[12] Yan L, Chunli Y. Characteristics and causes of accident in confined spaces[J]. China Safety Science Journal, 2017, 27(3): 141.
